# Antibiotic Management of Uncomplicated Skin and Soft Tissue Infections in the Real World

**DOI:** 10.3390/microorganisms11061369

**Published:** 2023-05-24

**Authors:** Luis Fernando Valladales-Restrepo, Brayan Stiven Aristizábal-Carmona, Jaime Andrés Giraldo-Correa, Luis Felipe Acevedo-Medina, Laura Valencia-Sánchez, Doménica Tatiana Acevedo-López, Andrés Gaviria-Mendoza, Manuel Enrique Machado-Duque, Jorge Enrique Machado-Alba

**Affiliations:** 1Grupo de Investigación en Farmacoepidemiología y Farmacovigilancia, Universidad Tecnológica de Pereira-Audifarma S. A, Pereira 660003, Colombia; lfvalladales@utp.edu.co (L.F.V.-R.); angaviria@utp.edu.co (A.G.-M.); memachado@utp.edu.co (M.E.M.-D.); 2Grupo de Investigación Biomedicina, Facultad de Medicina, Fundación Universitaria Autónoma de las Américas, Pereira 660003, Colombia; 3Semillero de Investigación en Farmacología Geriátrica, Grupo de Investigación Biomedicina, Facultad de Medicina, Fundación Universitaria Autónoma de las Américas, Pereira 660003, Colombia; brayan.aristizabal@uam.edu.co (B.S.A.-C.); jaime.giraldoc@uam.edu.co (J.A.G.-C.); luis.acevedo@uam.edu.co (L.F.A.-M.); laura.valencias@uam.edu.co (L.V.-S.); domenica.acevedo@uam.edu.co (D.T.A.-L.)

**Keywords:** abscess, antibacterial agents, cephalexin, inappropriate prescribing, pharmacoepidemiology, skin and tissue infection

## Abstract

*Background:* Skin and soft tissue infections are one of the main causes of consultations worldwide. The objective was to determine the treatment of a group of patients with uncomplicated skin and soft tissue infections in Colombia. *Methods:* Follow-up study of a cohort of patients with skin infections who were treated in the Colombian Health System. Sociodemographic, clinical and pharmacological variables were identified. Treatments were evaluated using clinical practice guidelines for skin infections. *Results:* A total of 400 patients were analyzed. They had a median age of 38.0 years and 52.3% were men. The most commonly used antibiotics were cephalexin (39.0%), dicloxacillin (28.0%) and clindamycin (18.0%). A total of 49.8% of the subjects received inappropriate antibiotics, especially those with purulent infections (82.0%). Being cared for in an outpatient clinic (OR: 2.09; 95% CI: 1.06–4.12), presenting pain (OR: 3.72; 95% CI: 1.41–9.78) and having a purulent infection (OR: 25.71; 95% CI: 14.52–45.52) were associated with a higher probability of receiving inappropriate antibiotics. *Conclusions:* Half of patients with uncomplicated skin and soft tissue infections were treated with antibiotics that were not recommended by clinical practice guidelines. This inappropriate use of antibiotics occurred in the vast majority of patients with purulent infections because the antimicrobials used had no effect on methicillin-resistant *Staphylococcus aureus*.

## 1. Introduction

Skin and soft tissue infections encompass a variety of pathological conditions involving the skin and subcutaneous tissue, fascia or muscle [1]. These infections can be simple and uncomplicated superficial (such as cellulitis, erysipelas and simple abscesses) or complicated deep (such as necrotizing fasciitis, infected ulcers and burns) [1]. Moreover, such infections can be nonpurulent (e.g., cellulitis and erysipelas) or purulent (e.g., abscesses, carbuncles or abscessed cellulitis) [2]. In uncomplicated infections, the main microorganism responsible for infections is *Streptococcus pyogenes,* with a lesser contribution from *Staphylococcus aureus,* while in complicated infections the main microorganism is *S. aureus* [3]. In Colombia, nonpurulent infections are mainly caused by *Streptococcus* spp. and methicillin-sensitive *Staphylococcus aureus* (MSSA), while purulent infections are mainly caused by methicillin-resistant *Staphylococcus aureus* (MRSA), *Streptococcus* spp. and anaerobes [4]. According to the SENTRY Antimicrobial Surveillance Program, 40.3% of the samples from 45 countries between 1997 and 2016 were MRSA [5]. In Colombia, in a multicenter study in patients with skin and soft tissue infections, the proportion of this microorganism was much higher (68.3%) [4].

Antimicrobial resistance is a global health and development threat. Antimicrobial-resistant organisms are found in people, animals, food, plants, and the environment. Such organisms can spread from person to person or between people and animals [6,7]. The inappropriate use of antibiotics is a public health problem [7]. It contributes to increasing antimicrobial resistance, hospital stays, higher health care costs and even mortality [7,8,9]. Empirical treatment of purulent skin and soft tissue infections should include coverage for MRSA [3]. For nonpurulent skin and soft tissue infections, there is no evidence that treating this microorganism improves clinical outcomes [3]. Therefore, the use of antibiotics such as cephalosporins (except the fifth generation) and penicillins is considered inappropriate for the management of purulent infections but is adequate for the management of nonpurulent infections [3,10]. In Colombia, between 2009 and 2016, 57% of patients with skin and soft tissue infections caused by MRSA received inadequate treatment [4]. For this reason, the clinical practice guidelines for the diagnosis and management of skin and soft tissue infections in Colombia was published in 2019 in an effort to improve and standardize the treatment of these patients [10].

The Colombian Health System offers universal coverage to the entire population through two affiliated regimes: the contributory regime that is paid by workers and employers and the subsidized regime that ensures everyone without the ability to pay and provides a significant number of specific antibiotics. Infections of the skin and soft tissues are one of the main causes of doctor visits worldwide, exceeded only by infections of the respiratory system and urinary tract [10]. It is a concern that there are not enough studies with real-world evidence on the management of these patients in Columbia [4,11] and no studies have been reported since the publication of clinical practice guidelines. Consequently, the present study aimed to determine the treatment outcomes of a group of patients with uncomplicated skin and soft tissue infections in Colombia.

## 2. Materials and Methods

### 2.1. Study Design and Patients

An observational, retrospective and follow-up study was carried out on a cohort of patients with skin and soft tissue infections identified from a population-based drug dispensing database that collects information on approximately 9.2 million people affiliated with the Colombia Health System. Patients affiliated with an insurer that serves approximately 3.8 million people distributed across most regions of the country were included in the study; 85% were affiliated with the contributory insurance system and 15% were affiliated with the government subsidy system.

The patients were identified using the codes of the International Classification of Diseases version 10 (ICD-10) and divided into groups with nonpurulent infections (cellulitis: L030-L033, L038, L039; erysipelas: A46X) and purulent infections (skin abscess: L020-L024, L028, L029) between 1 January and 31 December 2021. The first care visit was considered the index date for each patient and patients were followed for one year thereafter (31 December 2022). Patients aged 18 or older, of any sex and city of origin, were eligible. Patients without medical records who had changed insurance, had undergone organ transplants and had infections related to animal or human bites, diabetic foot, a surgical site, burns and immunosuppression (solid or hematological cancer, immunodeficiency virus) were excluded.

During the study period, a total of 21,077 people with skin and soft tissue infections were identified. A random sample of 377 patients was calculated using the Epi Info program and stratified according to the type of infection (purulent or nonpurulent), with an error of 5%, a confidence level of 95% and an expected frequency of 50%. The distribution of the sample was the same for the two strata.

### 2.2. Variables

Regarding the selected patients, we proceeded to review the electronic medical records that were consigned during the observation period and the first month of follow-up. From the information obtained, a database was designed that allowed the following groups of variables to be collected:Sociodemographic:

Sex, age, occupation, education and origin. The place of origin was categorized by departments according to the regions of Colombia, considering the classification of the National Administrative Department of Statistics (DANE) of Colombia, as follows: Bogotá-Cundinamarca region, Caribbean region, Central region, Eastern region, Pacific region and Amazon-Orinoquía region.

b.Clinics:

Signs/symptoms: pain, erythema/redness, induration, purulent discharge, fever, general malaise, among others.

Risk factors: use of antibiotics in the past month, hospitalization in the past month, dialysis replacement therapy and penetrating trauma.

Vital signs: systolic and diastolic blood pressure, heart rate, respiratory rate and temperature on admission.

Anthropometric measurements: weight, height and body mass index.

Diagnosis: purulent infection (abscesses, abscessed cellulitis, carbuncle), nonpurulent infection (cellulitis, erysipelas) and location (extremities, armpit, trunk, neck, head, among others).

Comorbidities: arterial hypertension, diabetes mellitus, dyslipidemia, hypothyroidism, ischemic heart disease, heart failure, chronic obstructive pulmonary disease, asthma, depression, anxiety, chronic kidney disease, among others.

c.Pharmacological/management:

Place of initial care: outpatient consultation, emergency service or home consultation; outpatient management vs. hospitalization.

Antibiotics: the following antibiotics were considered recommended according to the diagnosis [10].

Purulent infection: clindamycin, trimethoprim/sulfamethoxazole or linezolid are recommended in outpatient management; vancomycin, linezolid, daptomycin, clindamycin, tigecycline or ceftaroline are recommended in hospital management.

Nonpurulent infection: cephalexin, dicloxacillin, clindamycin, amoxicillin/clavulanate or trimethoprim/sulfamethoxazole are recommended in outpatient management; oxacillin, cefazolin, ampicillin/sulbactam or clindamycin are recommended in hospital management. Cellulitis associated with penetrating trauma, previous infection, intravenous drug use, abscessed cellulitis or immunosuppression should be managed as a purulent infection.

Inappropriate use of antibiotics was identified among patients who had received antibiotics other than those recommended.

d.Other management:

Incision and drainage for purulent infections, use of analgesics (acetaminophen, dipyrone, tramadol, codeine, morphine, nonsteroidal antiinflammatory drugs, ibuprofen, diclofenac, naproxen and celecoxib).

Pharmacological history in the past 90 days: subjects were grouped into the following categories: (a) antidiabetic drugs, (b) antihypertensive drugs and diuretics, (c) lipid-lowering drugs, (d) antiulcers, (e) antidepressants, (f) antipsychotics, (g) antihistamines and (h) others.

e.Follow-up:

Readmission/reconsultation: readmission of the patient within 30 days after discharge as a result of the same pathology; the reasons for readmission and management were determined.

New episodes of skin and soft tissue infections: number of purulent and nonpurulent infections per year from the index date.

### 2.3. Ethical Statement

The protocol was endorsed by the Bioethics Committee of the Technological University of Pereira in the category of “research without risk” (approval code: 07-140222) and by the ethics committee of the insurer. The principles of confidentiality of information established by the Declaration of Helsinki were respected.

### 2.4. Data Analysis

The data were analyzed with the statistical package SPSS Statistics, version 26.0 for Windows (IBM, Armonk, NY, USA). Descriptive analysis was performed with frequencies and proportions for the qualitative variables and measures of central tendency and dispersion for the quantitative variables through medians and interquartile ranges. The comparison of quantitative variables was performed using the Mann–Whitney U test and *X*^2^ or Fisher’s exact test for categorical variables. Multivariate binary logistic regression models were developed that included the variables in the bivariate analyses and variables that could be associated with inappropriate use of antibiotics (yes/no). The Hosmer–Lemeshow test was performed to describe the goodness of fit. The predictive capacity of the model was determined according to the area under the curve (AUC). The level of statistical significance was established at *p* < 0.05.

## 3. Results

### 3.1. Sociodemographic

A total of 400 patients in 33 different cities were analyzed, of whom 52.3% (*n* = 209) were men; the median age was 38.0 years (interquartile range: 29.0–50.8 years). A total of 52.5% (*n* = 210) were 18–39 years old, 40.0% (*n* = 160) were between 40–64 years old and 7.5% (*n* = 30) were 65 years or older. Most of the patients were in the Bogotá-Cundinamarca regions (*n* = 129; 32.3%) and the Caribbean (*n* = 129; 32.3%) and they had a secondary education (*n* = 124; 31.0%). The most frequent occupation was household activities (*n* = 73; 18.3%) and the most common affiliation scheme was contributory (*n* = 366; 91.5%). Table 1 describes the sociodemographic variables of the analyzed population.

### 3.2. Clinicians

Most of the patients presented with pain (*n* = 372, 93.0%) and erythema or redness (*n* = 324, 81.0%). The infections were mainly in the thigh or leg (*n* = 99; 24.8%), thorax or back (*n* = 55; 13.8%) and head or face (*n* = 41; 10.3%). A total of 58.0% (*n* = 232) had some chronic comorbidity, the most frequent being arterial hypertension (*n* = 79, 19.8%), obesity (*n* = 70, 17.5%) and dyslipidemia (*n* = 53; 13.3%). Table 2 describes the clinical variables of each group of patients.

### 3.3. Treatment

The patients were cared for mainly in outpatient services (*n* = 312; 78.0%) and emergency services (*n* = 83; 20.8%) (Table 3); most of them received outpatient management (*n* = 375; 93.8%). The most commonly used antibiotics were cephalexin (*n* = 156, 39.0%), dicloxacillin (*n* = 112, 28.0%), and clindamycin (*n* = 72, 18.0%) (Table 3 and Table 4). A total of 49.8% (*n* = 199) of the patients received inappropriate antibiotics, especially those with purulent infections (*n* = 164/200; 82.0%) and, to a lesser extent, those with nonpurulent infections (*n* = 35/200; 17.5%). Only 16.5% (*n* = 33/200) of the patients with purulent infections had evidence of incision and drainage in their medical histories. The majority received analgesics (*n* = 310; 77.5%). A total of 45.5% (*n* = 182) of the patients had received medication in the 90 days prior to the index date and, in the previous 30 days, 11.8% (*n* = 47) had received a prescription for an antibiotic. Table 3 describes the management received by patients with purulent and nonpurulent infections; Table 4 shows the pattern of the antibiotics used.

### 3.4. Multivariate Analysis

The binary logistic regression adjusted for sex, age, education, origin, affiliation regimen, symptoms, location of the infection, comorbidities and comedications revealed that being treated in an outpatient clinic, presenting pain and having a purulent infection were related to a higher probability of being prescribed an inappropriate antibiotic. No variable reduced this risk (Hosmer–Lemeshow test *p* = 0.995 and AUC = 0.862) (Table 5).

### 3.5. Follow-Up

A total of 14.0% (*n* = 56) of the patients were seen again after initial care; 89.3% (*n* = 50/56) complained of a lack of improvement or persistence of symptoms and 10.7% (*n* = 6/56) were referred for a check-up. A total of 82.1% (*n* = 46/56) were managed on an outpatient basis, while 17.9% (*n* = 10/56) required inpatient treatment. A total of 23.2% (13/56) of the cases had received inadequate management. At one year, 7.5% (*n* = 30/400) of the patients presented 1.13 (range 1–2) new episodes of skin and soft tissue infections, both purulent (*n* = 22/34; 64, 7%) and nonpurulent (*n* = 12/34; 35.3%). A total of 47.1% (16/34) had received inappropriate antibiotics.

## 4. Discussion

This study made it possible to identify the management received by a group of patients with purulent and nonpurulent infections of the skin and soft tissues and determined the adherence of the prescribers to the clinical practice guidelines. This evidence of the use of drugs in the real world may be useful for health care, academic and scientific personnel in making decisions regarding the risks faced by their patients and contribute to strengthening the rational use of antibiotics among physicians as a way to reduce antimicrobial resistance [12].

The median age of the patients was lower than that found in other studies (38.0 vs. 41.1–61.6 years) [4,13,14,15,16,17], with a predominance of men, similar to previous studies (52.3% vs. 50.7–63.0%) [4,13,14,15,16,17,18,19]. The clinical manifestations of patients with skin and soft tissue infections, as well as the location of infections predominantly in the lower limbs, were consistent with those reported in the literature [13,15,16,19,20]. Hypertension was the most prevalent comorbidity in this report, consistent with what was published in a multicenter study in six European countries, although in a lower proportion (19.8% vs. 48.4%) [14], and contrasts with other reports where diabetes mellitus predominated (10.0% vs. 18.5%–51.0%) [4,13,15,18,19,20].

Half of the patients received antibiotics not recommended by the clinical practice guidelines. In Colombia, several studies on antibiotics in the outpatient setting have found a high proportion of inappropriate prescriptions [11,21,22,23]. Cephalosporins have been used to treat conditions not indicated in 55.6% of cases [11], macrolides in 31.3% of cases [21], fluoroquinolones in 24.0% of cases [23], and tetracyclines in 23.5% of cases [22]. This problem has also been described in international studies [12,24]. In a systematic review and meta-analysis, the inappropriate use of antibiotics in the outpatient setting was found in 8.0% to 100% of the records [12]. In another systematic review and meta-analysis of hospitalized patients, the rate of inappropriate use was between 14.1% and 78.9% [24]. The improper and excessive use of antimicrobials is the main factor that determines the appearance of drug-resistant pathogens [7].

Cephalexin and dicloxacillin were the most widely used antibiotics, which is consistent with a study of antibiotic-use patterns in Colombia [25]. These antibiotics are clearly indicated in uncomplicated nonpurulent infections because Gram-positive cocci such as *S. pyogenes* and MSSA prevail [2,10]. However, the proportion of patients who received them inappropriately for the management of purulent infections or who had risk factors for MRSA was high. For this group of patients, clindamycin and trimethoprim/sulfamethoxazole were among the recommended antibiotics, but these drugs were used in less than a fifth of cases. This contrasts with a study conducted in the United States in which trimethoprim/sulfamethoxazole was used in 50.4% of patients and clindamycin in 16.3% [26].

Some variables were found to be related to an increased probability of receiving antibiotics inappropriately; patients with purulent infections had a 25-times higher risk than those with nonpurulent infections. In this group of patients, four out of five patients received inappropriate antibiotics. This finding is consistent with other reports [11,17,20]. In Canada, Ibrahim et al. found that 63.3% of patients did not receive the antimicrobials recommended by management guidelines [17], while, in the United States, Sutton et al. found that among patients who required hospital management, 73.0% did not receive adequate antibiotic management [20]. In Colombia, Gaviria-Mendoza et al. found that 46.2% of patients with purulent infections were improperly managed with cephalosporins [11]. The Colombian guidelines recommend that the empirical management of uncomplicated purulent infections should involve an antibiotic that covers MRSA; therefore, the use of penicillins, cephalosporins (except those of the fifth generation), macrolides, tetracyclines, aminoglycosides or fluoroquinolones are considered inappropriate treatments [10]. The Infectious Diseases Society of America (Arlington, VA, USA) guide considers the use of tetracyclines appropriate [2]; similarly, they emphasize the importance of drainage and incision of infections larger than 2 cm [2,10]. However, less than a fifth of the patients had a record of the procedure, which is consistent with the findings of a Canadian publication [18].

This study revealed that patients seen in the outpatient clinic were more likely to inappropriately receive antibiotics. This finding is consistent with a multicenter study conducted in Korea on the prescription of antibiotics; inappropriate use was observed more frequently in outpatient care than in emergency room visits (30.9% vs. 20.4%; *p* < 0.001) [27]. However, it contrasts with a study in China, where the proportion of inappropriate use of antibiotics was higher in emergency services than in outpatient services (62.0% vs. 48.4%) [28]. Additionally, in this report, it was identified that patients who presented pain were also more likely to inappropriately receive antibiotics. This is in line with what was described in an investigation on the use of antibiotics in patients with animal bites, in which it was shown that those who were prescribed analgesics had a higher risk of receiving an antibiotic not recommended by the guidelines [29].

In this cohort, one out of every seven patients was seen again by medical professionals in the first 30 days after the initial visit, which is very similar to what was found in Canada (13.2%) [17] and the United States (13.0%) [20]. A total of 8.0% of the patients had a new skin and soft tissue infection after one year of follow-up, although in other studies, the proportion was higher [19,30,31]. In the United States, recurrent *S. aureus* infections of the skin and soft tissues have been reported to be common among healthy adults and occur in at least one in six people during the first year after the index event [30]. In Spain, readmission at 6 months occurred in 12.3–26.3% of patients who required hospital management [19,31]. The management of these cases involved identifying and treating predisposing conditions (for example, edema, eczema, venous insufficiency, obesity, among others) and considering the administration of prophylactic antibiotics or the use of decolonization regimens [2].

Some limitations should be considered when interpreting the results of this study. The data were obtained from a group of patients mainly from the contributory regime of the Colombian health system, so the findings may not be extrapolated to patients in different insurance settings. In addition, for some variables, information was not available for the total number of patients because such information was missing from the medical records. The patients had uncomplicated infections, so there were no microbiological studies to evaluate the etiological agents and the patterns of sensitivity or resistance. However, a significant number of patients distributed throughout most of the geographic regions of the national territory, involving both purulent and nonpurulent infections, were included.

## 5. Conclusions

With these findings, it was concluded that half of the patients with uncomplicated skin and soft tissue infections were treated with antibiotics that were not recommended by clinical practice guidelines. This inappropriate use of antibiotics was present in the vast majority of purulent infection cases because the antimicrobials used had no effect on MRSA. In addition, a low proportion of patients underwent incision and drainage of the purulent collections. The Ministry of Health and Social Protection, as well as the different scientific societies in Columbia, should implement strategies so that clinical practice guidelines are more visible to prescribing doctors and more continuing education programs should be promoted.

## Figures and Tables

**Table 1 microorganisms-11-01369-t001:** Sociodemographic variables of a group of patients with skin and soft tissue infections, Colombia.

Variables	Purulent Infections		Nonpurulent Infections		*p*
	*n* = 200	%	*n* = 200	%	
Men	98	49.0	111	55.5	0.193
Age, median (IQR)	36.0 (28.0–47.0)		40.0 (30.0–53.0)		0.023 ^a^
Origin	-	-	-	-	
Bogotá-Cundinamarca Region	46	23.0	83	41.5	<0.001
Caribbean Region	73	36.5	56	28.0	0.069
Central Region	62	31.0	42	21.0	0.023
Pacific Region	17	8.5	8	4.0	0.063
Eastern Region/Orinoquia-Amazonia	2	1.0	11	5.5	0.020 ^b^
Scholarship	-	-	-	-	
Primary	10	5.0	13	6.5	0.519
Secondary	74	37.0	50	25.0	0.009
University	32	16.0	15	7.5	0.008
Occupation	-	-	-	-	
Household activities	36	18.0	37	18.5	0.897
Student	19	9.5	12	6.0	0.191
Worker	13	6.5	16	8.0	0.563
Affiliation regime	-	-	-	-	
Contributory	183	91.5	183	91.5	1.000
Subsidized	17	8.5	17	8.5	

IQR—interquartile range, ^a^ Mann–Whitney U test, ^b^ Fisher’s exact test.

**Table 2 microorganisms-11-01369-t002:** Clinical variables of a group of patients with skin and soft tissue infections, Colombia.

Variables	Purulent Infections		Nonpurulent Infections		*p*
	*n* = 200	%	*n* = 200	%	
Signs/Symptoms	-	-	-	-	
Pain	180	90.0	192	96.0	0.019
Erythema or flushing	143	71.5	181	90.5	<0.001
Induration	122	61.0	0	0.0	<0.001
Fever	17	8.5	23	11.5	0.317
General discomfort	10	5.0	25	12.5	0.008
Risk factors	-	-	-	-	
Previous use of antibiotics	26	13.0	21	10.5	0.438
Penetrating trauma	2	1.0	5	2.5	0.449 ^a^
Recent hospitalization	0	0.0	2	1.0	0.499 ^a^
Renal replacement therapy	0	0.0	1	0.5	1.000 ^a^
Vital signs	-	-	-	-	
Systolic blood pressure (mmHg), median (IQR)	110.0 (110.0–116.0)		110.0 (110.0–120.0)		0.012 ^b^
Diastolic blood pressure (mmHg), median (IQR)	70.0 (70.0–75.0)		70.0 (70.0–77.8)		0.016 ^b^
Heart rate (beats/minute), median (IQR)	78.0 (75.0–84.0)		78.0 (75.0–83.0)		0.481 ^b^
Respiratory rate (breaths/minute), median (IQR)	18.0 (16.0–19.0)		18.0 (17.0–19.0)		0.231 ^b^
Temperature (°C), median (RIQ)	36.2 (36.0–36.5)		36.0 (36.0–36.5)		0.524 ^b^
Anthropometric measures	-	-	-	-	
Weight (kg), median (IQR)	70.0 (60.0–79.0)		70.0 (60.0–80.0)		0.757 ^b^
Body mass index (kg/m^2^), median (IQR)	25.4 (22.5–28.7)		25.7 (23.7–28.3)		0.384 ^b^
Location of infection	-	-	-	-	
Thigh or leg	40	20.0	59	29.5	0.028
Thorax or back	35	17.5	20	10.0	0.029
Head or face	16	8.0	25	12.5	0.138
Gluteus	26	13.0	9	4.5	0.003
Forearm or arm	14	7.0	15	7.5	0.847
Comorbidities	-	-	-	-	
Arterial hypertension	38	19.0	41	20.5	0.706
Obesity	35	17.5	35	17.5	1.000
Dyslipidemia	25	12.5	28	14.0	0.658
Diabetes mellitus	21	10.5	19	9.5	0.739
Migraine	13	6.5	19	9.5	0.269

IQR—interquartile range, ^a^ Mann–Whitney U test, ^b^ Fisher’s exact test.

**Table 3 microorganisms-11-01369-t003:** Pharmacological management of a group of patients with infections of the skin and soft tissues, Colombia.

Variables	Purulent Infections		Nonpurulent Infections		*p*
	*n* = 200	%	*n* = 200	%	
Place of care	-	-	-	-	
External consultation	169	84.5	143	71.5	0.002
Emergencies	31	15.5	52	26.0	0.010
Home consultation	0	0.0	5	2.5	0.061 ^a^
Antibiotics	-	-	-	-	
Cephalexin	81	40.5	75	37.5	0.539
Dicloxacillin	56	28.0	56	28.0	1.000
Clindamycin	29	14.5	43	21.5	0.068
Gentamicin	14	7.0	8	4.0	0.188
Trimethoprim/sulfamethoxazole	8	4.0	8	4.0	1.000
Analgesics	-	-	-	-	
Naproxen	96	48.0	105	52.5	0.368
Acetaminophen	25	12.5	40	20.0	0.042
Diclofenac	28	14.0	32	16.0	0.575
Ibuprofen	19	9.5	15	7.5	0.473
Tramadol	9	4.5	5	2.5	0.276
Comedications	-	-	-	-	
Analgesics/antiinflammatories	55	27.5	68	34.0	0.159
Antihypertensives and diuretics	24	12.0	26	13.0	0.762
Lipid-lowering drugs	18	9.0	24	12.0	0.328
Antiulcer medication	18	9.0	15	7.5	0.586
Antidiabetics	13	6.5	13	6.5	1.000

^a^ Fisher’s exact test.

**Table 4 microorganisms-11-01369-t004:** Pattern of antibiotic use, frequency of use, prescribed dose, days of treatment, distribution by sex, age and appropriate use in a group of patients with infections of the skin and soft tissues, Colombia.

Antibiotic	*n* = 400	%	Prescribed Dose (mg/day)			Days	Sex		Age	Appropriate Use	
			Median (IQR)	Mode	nDDD ^a^	Median (IQR)	M (%)	F (%)	Median (IQR)	Yes (%)	No (%)
Cephalexin	156	39.0	2000 (1500–2000)	2000	0.9	7 (7–7)	51.3	48.7	37.0 (29.3–49.8)	45.5	54.5
Dicloxacillin	112	28.0	2000 (1750–2000)	2000	0.9	7 (7–7)	59.8	40.2	38.5 (27.0–48.0)	42.0	58.0
Clindamycin	72	18.0	1200 (900–1800)	1800	1.2	7 (4–7)	48.6	51.4	40.5 (30.0–52.8)	100.0	0.0
Gentamicin	22	5.5	160 (157.5–160)	160	0.6	3 (3–4.3)	63.6	36.4	41.0 (34.0–47.8)	0.0	100.0
Trimethoprim/sulfamethoxazole	16	4.0	400 (320–480)	480	1.0	7 (7–10)	43.8	56.3	41.0 (31.0–47.0)	100.0	0.0
Doxycycline	14	3.5	200 (200–225)	200	2.1	7 (7–15)	14.3	85.7	34.5 (24.8–44.0)	0.0	100.0
Cephradine	8	2.0	1500 (1000–2000)	1000	-	7 (7–7)	50.0	50.0	53.5 (36.5–68.8)	12.5	87.5
Ciprofloxacin	7	1.8	800 (800–800)	800	1.0	7 (7–10)	57.1	42.9	48.0 (35.0–63.0)	0.0	100.0
Amoxicillin	5	1.3	1500 (1500–1750)	1500	1.1	7 (7–7)	80.0	20.0	49.0 (26.5–54.5)	0.0	100.0
Amoxicillin/clavulanate	4	1.0	2000 (1906.3–2750.0)	2000	1.5	7 (7–7)	50.0	50.0	37.0 (29.8–43.5)	50.0	50.0
Metronidazole	4	1.0	1500 (1125.0–1500)	1500	0.9	7 (2.5–7)	50.0	50.0	36.0 (26.5–47.8)	0.0	100.0
Oxacillin	4	1.0	2000 (2000–6500)	2000	1.8	3 (1.5–3.8)	25.0	75.0	40.0 (23.3–42.5)	75.0	25.0
Cefazolin	3	0.8	2000 (1500–3000)	1500	0.7	7 (7–8)	33.3	66.7	40.0 (27.0–57.0)	100.0	0.0
Ceftriaxone	3	0.8	2000 (1000–2000)	2000	0.8	3 (1–4)	0.0	100.0	68.0 (26.0–72.0)	0.0	100.0
Benzathine benzylpenicillin	3	0.8	24,00,000 (1,200,000–2,400,000)	2,400,000	-	1 (1–1)	66.7	33.3	30.0 (28.0–58.0)	0.0	100.0
Amikacin	2	0.5	500 (500–500)	500	0.5	4.5 (4–5)	0.0	100.0	35.5 (25.0–46.0)	0.0	100.0
Cephalothin	2	0.5	2000 (1000–3000)	1000	0.5	4 (1–7)	50.0	50.0	58.0 (39.0–77.0)	0.0	100.0
Cefepime	1	0.3	4000	4000	1.0	7	100.0	0.0	82.0	0.0	100.0
Clarithromycin	1	0.3	1000	1000	2.0	3	0.0	100.0	37.0	0.0	100.0
Norfloxacin	1	0.3	800	800	1.0	7	100.0	0.0	21.0	0.0	100.0
Vancomycin	1	0.3	2000	2000	1.0	9	100.0	0.0	26.0	100.0	0.0

IQR—interquartile range, M—male, F—female, ^a^ proportion between the daily dose received and the defined daily dose.

**Table 5 microorganisms-11-01369-t005:** Binary logistic regression of variables related to receiving inappropriate antibiotics in initial care in a group of patients with infections of the skin and soft tissues, Colombia.

Variables	Sig.	OR	95%CI	
			Lower	Upper
Men	0.285	1.353	0.777	2.354
Age < 40 years	0.747	1.100	0.616	1.966
Caribbean region	0.654	0.874	0.486	1.574
Scholarship university	0.570	1.278	0.548	2.980
Contributory regime	0.276	1.721	0.648	4.567
Arterial hypertension	0.922	1.037	0.504	2.131
Obesity	0.764	0.894	0.431	1.854
Outpatient care	0.033	2.093	1.063	4.123
Presence of pain	0.008	3.721	1.415	9.785
Thigh/leg location	0.573	1.196	0.642	2.227
Purulent infection	<0.001	25.716	14.527	45.522
Receipt of medication in the last 3 months	0.538	1.199	0.673	2.136

Sig—statistical significance, OR—odds ratio, CI—confidence interval.

## Data Availability

https://www.protocols.io/private/B906992FE07811EDA84B0A58A9FEAC02. Access on 21 April 2023.

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
