# Peer review of "Antibiotic Management of Uncomplicated Skin and Soft Tissue Infections in the Real World"

_microorganisms, 2023, doi:10.3390/microorganisms11061369_

Round 1
Reviewer 1 Report
This article demonstrated the antibiotic management of uncomplicated skin and soft tissue infections in the real world which was considered to contribute to the success of the therapy. The experimental design and the results seem to be exquisite and valid, however, the following points would be considered to improve this article.
1. Lines 42, 48 and 54, methicillin-resistant Staphylococcus aureus words methicillin-resistant rewritten without italics.
2. Lines 61 and 62, Infections of the skin and soft tissues are one of the main causes of doctor visits worldwide, exceeded only by infections of the respiratory system and urinary tract. Delete space between the words visits and worldwide.
3. The authors do not mention anything about the types of microorganisms that cause purulent and nonpurulent infections or their relationships with the appropriate and inappropriate use of antibiotics.
4. Lines 257 and 259, methicillin-sensitive Staphylococcus aureus and methicillin-resistant Staphylococcus aureus words methicillin-sensitive and methicillin-resistant rewritten without italics
5. Lines 279 and 280, However, less than a fifth of the patients had a record of the procedure, which is consistent with the findings of a Canadian publication [17]. Delete space between the words record of and the procedure.
Author Response
Response to reviewer 1
Manuscript ID: microorganisms-2386685
Type: Article
Title: Antibiotic Management of Uncomplicated Skin and Soft Tissue Infections in the Real World
Review Report Form
Open Review
( ) I would not like to sign my review report
(x) I would like to sign my review report
Quality of English Language
( ) I am not qualified to assess the quality of English in this paper
( ) English very difficult to understand/incomprehensible
( ) Extensive editing of English language required
( ) Moderate editing of English language
( ) Minor editing of English language required
(x) English language fine. No issues detected
|
Yes |
Can be improved |
Must be improved |
Not applicable |
|
|
Does the introduction provide sufficient background and include all relevant references? |
(x) |
( ) |
( ) |
( ) |
|
Are all the cited references relevant to the research? |
(x) |
( ) |
( ) |
( ) |
|
Is the research design appropriate? |
(x) |
( ) |
( ) |
( ) |
|
Are the methods adequately described? |
(x) |
( ) |
( ) |
( ) |
|
Are the results clearly presented? |
(x) |
( ) |
( ) |
( ) |
|
Are the conclusions supported by the results? |
(x) |
( ) |
( ) |
( ) |
Comments and Suggestions for Authors
This article demonstrated the antibiotic management of uncomplicated skin and soft tissue infections in the real world which was considered to contribute to the success of the therapy. The experimental design and the results seem to be exquisite and valid, however, the following points would be considered to improve this article.
- Lines 42, 48 and 54, methicillin-resistant Staphylococcus aureuswords methicillin-resistant rewritten without italics.
Answer: The change is made. And at the suggestion of the second evaluator, the initials are placed.
- Lines 61 and 62, Infections of the skin and soft tissues are one of the main causes of doctor visits worldwide, exceeded only by infections of the respiratory system and urinary tract. Delete space between the words visits and worldwide.
Answer: The change is made.
- The authors do not mention anything about the types of microorganisms that cause purulent and nonpurulent infections or their relationships with the appropriate and inappropriate use of antibiotics.
Answer: The change is made. Information is added in the introduction.
- Lines 257 and 259, methicillin-sensitive Staphylococcus aureusand methicillin-resistant Staphylococcus aureus words methicillin-sensitive and methicillin-resistant rewritten without italics
Answer: The change is made.
- Lines 279 and 280, However, less than a fifth of the patients had a record of the procedure, which is consistent with the findings of a Canadian publication [17]. Delete the space between the words record of and the procedure.
Answer: The change is made.
Reviewer 2 Report
The manuscript by Valladales-Restrepo et al. is original and add data in the knowledge of the treatment outcomes of a group of patients with uncomplicated skin and soft tissue infections in Colombia. I support its further processing after appropriate modifications as outlined below.
L33-36: please divide this sentence, presently is too long and hardly undersatndable
L37: delete „they can be”
L48: „methicillin-resistant Staphylococcus aureus” – after its first appearence in the text, please use the abbreviated form (e.g. MRSA)
L40-41: within the Introduction section the authors must highlight the importance and also the challanges raised by of staphylococcal flora in skin infections for the veterinary medicine consulting and citing some scientific papers (e.g. doi: 10.1136/vr.101426 and doi: 10.3390/microorganisms9030515)
L94-L 146: the presentation of subheadings 2.1, 2.2, 2.3, 2.4 and 2.5 (except the statistically interpretation of data) are not scientifically sound, and hardly understandable. Please try to find the way to improve them!
L315: „we can” – please rephrase avoiding the personal mode formulations, it is not so characteristic for the scientific style.
I strongly recommend to the authors to carefully revise the English content of the manuscript, either through a native English speaker colleague, or via a professional English editing service. There are a lot of hardly understandable sentences (e.g. L95, L101-102, etc.).
I strongly recommend to the authors to carefully revise the English content of the manuscript, either through a native English speaker colleague, or via a professional English editing service. There are a lot of hardly understandable sentences (e.g. L95, L101-102, etc.).
Author Response
Response to reviewer 2
Manuscript ID: microorganisms-2386685
Type: Article
Title: Antibiotic Management of Uncomplicated Skin and Soft Tissue Infections in the Real World
Review Report Form 2
Open Review
(x) I would not like to sign my review report
() I would like to sign my review report
Quality of English Language
() I am not qualified to assess the quality of English in this paper
() English very difficult to understand/incomprehensible
() Extensive editing of English language required
(x) Moderate editing of English language
() Minor editing of English language required
() English language fine. No issues detected
|
Yes |
Can be improved |
Must be improved |
Not applicable |
|
|
Does the introduction provide sufficient background and include all relevant references? |
() |
(x) |
() |
() |
|
Are all the cited references relevant to the research? |
() |
(x) |
() |
() |
|
Is the research design appropriate? |
(x) |
() |
() |
() |
|
Are the methods adequately described? |
() |
() |
(x) |
() |
|
Are the results clearly presented? |
(x) |
() |
() |
() |
|
Are the conclusions supported by the results? |
() |
(x) |
() |
() |
Comments and Suggestions for Authors
The manuscript by Valladales-Restrepo et al. is original and add data in the knowledge of the treatment outcomes of a group of patients with uncomplicated skin and soft tissue infections in Colombia. I support its further processing after appropriate modifications as outlined below.
L33-36: please divide this sentence, presently is too long and hardly undersatndable
Answer: The change has been made.
L37: delete "they can be”
Answer: The change has been made.
L48: „methicillin-resistant Staphylococcus aureus” – after its first appearence in the text, please use the abbreviated form (e.g. MRSA)
Answer: The change has been made.
L40-41: within the Introduction section the authors must highlight the importance and also the challanges raised by of staphylococcal flora in skin infections for the veterinary medicine consulting and citing some scientific papers (e.g. doi: 10.1136/vr.101426 and doi: 10.3390/microorganisms9030515)
Answer: A comment has been added to the introduction. Two new references are introduced.
L94-L 146: the presentation of subheadings 2.1, 2.2, 2.3, 2.4 and 2.5 (except the statistically interpretation of data) are not scientifically sound, and hardly understandable. Please try to find the way to improve them!
Answer: We have numbered the headings for ease of understanding. For each subtitle, the sociodemographic, clinical, pharmacological, follow-up variables and the ethical statement are described to facilitate understanding.
L315: „we can” – please rephrase avoiding the personal mode formulations, it is not so characteristic for the scientific style.
Answer: The change has been made.
Comments on the Quality of English Language
I strongly recommend to the authors to carefully revise the English content of the manuscript, either through a native English speaker colleague or via a professional English editing service. There are a lot of hardly understandable sentences (e.g. L95, L101-102, etc.
Answer: The manuscript was translate and edited by American Journal Experts. We add the certificate.
Round 2
Reviewer 2 Report
Much improved!